# Lipid and Protein Oxidation of Brown Rice and Selenium-Rich Brown Rice during Storage

**DOI:** 10.3390/foods11233878

**Published:** 2022-12-01

**Authors:** Minghui Zhang, Kunlun Liu

**Affiliations:** 1College of Food Science and Engineering, Henan University of Technology, Zhengzhou 450001, China; 2School of Food and Reserves Storage, Henan University of Technology, Zhengzhou 450001, China

**Keywords:** storage protein, proteomics, metabolic pathway, sulfur metabolism, oxidation resistance

## Abstract

Selenium-rich rice has become one of the effective ways to increase people’s selenium intake. Selenium-containing proteins have higher antioxidant properties, which may lead to selenium-rich brown rice (Se-BR) having better storage stability than ordinary brown rice (BR). By measuring the peroxidation value, fatty acid value, carbonyl value and protein secondary structure, it was found that Se-BR had higher oxidation resistance stability than BR. The biological function of the differential proteins (DEPs) between ordinary brown rice stored for 0 days (BR-0) and 180 days (BR-6) as well as Se-rich brown rice stored for 0 days (Se-0) and 180 days (Se-6) was investigated by using iTRAQ. A total of 237, 235, 113 and 213 DEPs were identified from group A (BR-0/BR-6), group B (Se-0/Se-6), group C (BR-0/Se-0) and group D (BR-6/Se-6), respectively. Kyoto Encyclopedia of Genes and Genomes analysis showed that the DEPs were mainly enriched in glucose metabolism, tricarboxylic acid cycle, fatty acid biosynthesis and degradation, glutathione metabolism, sulfur metabolism, peroxisome and other metabolic pathways. This study provides theoretical support for the study of protein oxidation kinetics and storage quality control of brown rice during storage.

## 1. Introduction

Selenium is an essential trace element for the human body and is closely related to human health. It has significant effects on antioxidation [1,2], the prevention of cardiovascular and cerebrovascular diseases, the improvement of body immunity [3] and antitumor and antiaging properties [4]. The human body mainly supplements selenium through diet. Two thirds of the world’s soil is deficient in selenium, resulting in insufficient selenium intake [5]. Selenium is mainly involved in the regulation of physiological functions in the form of proteins in the human body. Rotruck et al. [6] proposed in 1973 that selenium participates in the formation of glutathione peroxidase (GPx). With further research, selenium has been proved to replace sulfur to form selenocysteine (SeCys), which exists at the GPx active center. Moreover, it is the cofactor of GPx, thioredoxin reductase and other biological enzymes [7].

Among the three major grains in the world, rice is the main grain supply in China. About 50% of people in the world regard rice as the main source of energy, and it is widely used in food industry production [8]. China’s rice production ranks first in the world, accounting for about 30% of the world production.

The organic selenium in Se-rich rice is mainly present in the form of selenomethionine (SeMet), selenocysteine (SeCys_2_) and methylselenocysteine (MetSeCys) [1,9]. The characteristics of rice production and consumption indicate that rice needs to be stored for a certain period of time. Owing to the long-term storage process, unstable factors, such as temperature and humidity, may affect the quality of rice after storage. Lipids and proteins in rice undergo complex physical and chemical changes during storage [10]. Studies have shown that selenium-enriched rice has stronger antioxidant activity than ordinary rice [2,11], which may be related to selenium-substituted proteins participating in different metabolic pathways of proteins.

The lipid content of brown rice accounts for 1–2%, but during storage, unsaturated fatty acids are easily oxidized and decomposed, producing peroxidized free radicals and other secondary oxidation products. Free radicals, as a potential initiating factor, initiate chain polymerization in the protein reaction system through hydrogen extraction [12], which promotes complex changes in proteins and aging and deterioration of rice while affecting the edible quality and nutritional value of rice [13]. The protein content of brown rice accounts for 8–12%. Similar to the process of lipid oxidation, protein/Se-protein oxidation is affected by environmental factors and reactive oxygen species in grains (such as O_2_^−^, OH, H_2_O_2_, alkoxy radicals and lipid free radicals) during storage, resulting in intramolecular or intermolecular cross-linking of proteins and affecting the structure, nutrition and function of proteins [14]. Methionine (Met), tyrosine, histidine, phenylalanine, tryptophan, lysine and arginine are sensitive to the oxidation of proteins, and they are important targets for lipid-induced oxidation [15,16]. Some studies have shown that the solubility, foam stability, gelation and thermal stability of proteins increase after being subjected to appropriate oxidative attack, which may be related to the proper development of protein structure, indicating that appropriate oxidation can improve the functional properties of proteins in some aspects, which has positive significance for improving food quality. However, excessive oxidation will seriously break the structure of protein, leading to the decline of the protein’s functional properties, flavor deterioration and nutritional loss, and even produce toxic and harmful substances. The current research on the interaction between lipid oxidation products and proteins has involved the formation and transfer of lipid free radicals, the aggregation of protein molecules induced by lipid oxidation products and the change of functional properties [17,18,19].

Proteomics is a research technique that finds certain specific protein molecules between samples by studying protein characteristics on a large scale, including protein expression, translation modification and protein–protein interaction [20]. Proteomics has been widely used in many fields to reveal the structure, abundance and interactions between proteins. Wu et al. [21] analyzed the mechanism of gastric cancer metastasis promoted by deubiquitinating enzyme ubiquitin-specificprotease3 via the isobaric tags for relative and absolute quantitation (iTRAQ) technique. Yang et al. [22] identified differentially expressed proteins (DEPs) in the early grain development of large-grain and small-grain wheat cultivars by using proteomic methods and elucidated the molecular and cellular processes leading to these differences.

At present, there are few reports on the changes in the protein oxidation mechanism and the functional properties of Se-rich rice during storage. The purpose of this study is to analyze the changes in lipids and proteins in Se-rich brown rice and ordinary brown rice after storage under the same storage conditions for a certain period of time. The mechanism of quality deterioration of the two kinds of rice during storage is also clarified from the level of proteomics.

## 2. Materials and Methods

### 2.1. Sample

Ordinary brown rice (BR) and Se-rich brown rice (Se-BR) were obtained from Yuanyang Basu Rice Industry Co., LTD (Xinxiang, China) in 2021. The rice variety is Huangjinqing 22438, which is commonly cultivated in China. The moisture content of BR and Se-BR was 12.99% and 13.20%, respectively.

### 2.2. Storage Conditions

Two kinds of samples of brown rice were packed in clean cloth bags and placed in a constant temperature and humidity incubator at 35 °C and 75% RH for 180 days. Samples were taken every 30 days for further analysis.

### 2.3. Preparation of Brown Rice Oil

The brown rice powder was extracted in hexane for 12–16 h at room temperature. The filtered supernatant was evaporated in a water bath to obtain brown rice oil, which was stored at 4 °C.

### 2.4. Preparation of Brown Rice Protein

Brown rice protein was extracted according to method reported by Liu et al. [2]. Ordinary brown rice protein (BRP) and selenium-rich brown rice protein (Se-BRP) were extracted with deionized water at pH 9.0. The mixture was rotated at 45 °C for 3 h and then centrifuged at 4000× *g* for 20 min. The supernatant was adjusted to pH 4.5 and then centrifuged at 5000× *g* for 10 min. Protein precipitation was obtained after the supernatant was removed. After freeze-drying, the protein was stored at −20 °C.

### 2.5. Determination of Peroxide Value (PV)

The PV was determined according to method reported by Mehta et al. [23]. Briefly, 0.2 g (accurate to 0.01) of sample was taken into a 15 mL centrifuge tube. Then, 9.8 mL of chloroform-methanol (7/3, *v*/*v*) solution was added, and the mixture was oscillated for 5 s. Next, 100 µL of xylenol orange solution (10 mmol/L) and 50 µL of FeCl_2_ solution (36 mmol/L) were added, and the absorbance at 560 nm was measured after a 5-min reaction.

### 2.6. Determination of Fatty Acid Value (FAV)

The FAV was determined according to method reported by Shu et al. [24]. Herein, 10 g of rice powder was extracted with benzene for 30 min. Subsequently, 0.04% phenolphthalein ethanol was added into 25 mL of filtrate and was titrated with 0.01 mol/L potassium hydroxide.

### 2.7. Determination of Carbonyl Value (CV)

The CV was determined according to method reported by Hasan et al. [25]. For this, 6 mL of 2,4-dinitrophenylhydrazine (DNPH, 10 mmol/L) solution was used to react with 2.1 mL of protein solution (3.5 g/mL) to form a brownish-red precipitate. Next, 2.7 mL of trichloroacetic acid (40%) was added to react for 20 min. After centrifugation, 9 mL of ethyl acetate and ethanol solution (1:1, *v*/*v*) were used to wash the precipitate 3 times. The precipitate was dissolved with 6 mL of guanidine hydrochloride (6 mol/L). The absorbance at 367 nm of the solution was determined and the extinction coefficient was 22,000 M^−1^ cm^−1^.

### 2.8. Fourier-Transformed Infrared (FTIR) Spectroscopy

Samples were mixed with potassium bromide in a ratio of 1:100 and scanned 32 times in the wavenumber range of 4000–400 cm^−1^ by the Nicolet6700 Fourier-Transform Infrared Spectrometer (Waltham, MA, USA). PeakFit (version 4.12, Palo Alto, CA, USA) software was used to analyze the infrared spectra of amide I bands (1600–1700 cm^−1^) of protein. The relative content of the secondary structure was calculated [20].

### 2.9. Proteomic Analysis

#### 2.9.1. Protein Extraction

Brown rice protein was extracted by the method of Wang et al. [26]. The treated protein clumps were stored at −80 °C for further use.

#### 2.9.2. Protein Quantification

Lysate L3 was added to dissolve the protein, and ultrasonic treatment-assisted solubilization was performed after freezing centrifugation. The BSA was taken to make a standard curve.

#### 2.9.3. Protein Filter-Aided Sample Preparation (FASP) Enzymatic Lysis

The protein solution (200 µg) was kept in a centrifuge tube, and 4 μL of TCEP reducing reagent (60 °C for 1 h) and 2 μL of MMTS cysteine (Cys)-blocking reagent (room-temperature reaction for 30 min) were added subsequently. The reduced alkylated protein solution was added to a ultrafiltration tube (10 kDa) and centrifuged (12,000× *g* at 4 °C for 20 min), and the bottom solution was discarded. Likewise, 100 μL of 8 mol/L urea (pH 8.5) was added and centrifuged, and the bottom solution was discarded; this process was repeated 2 times. Afterward, 100 μL of 0.25 mol/L TEAB (pH 8.5) was added and centrifuged, and the bottom solution was discarded; this process was repeated 3 times. The collection tube was then replaced. TEAB (50 μL of 0.5 mol/L) was added to an ultrafiltration tube, followed by trypsin (1:50 trypsin/protein, 37 °C reaction overnight). The next day, trypsin (1:100 trypsin/protein, 37 °C for 4 h), was added and centrifuged (12,000× *g* for 20 min). Subsequently, 50 μL of 0.5 mol/L TEAB was added to the ultrafiltration tube and centrifuged (12,000× *g* at 4 °C for 20 min), and this step was merged with the previous step. Finally, a total of 100 μL of the enzymatic digested sample at the bottom of the tube was collected.

#### 2.9.4. Labeling and Analysis of iTRAQ Reagent

The samples were labeled with a 6-standard iTRAQ kit and preserved for use after vacuum freeze-centrifugal drying.

Liquid chromatography–mass spectrometry (LC-MS) was performed according to Zhou et al. [27]. The mass spectrum parameters were as follows: (1) primary mass spectrometry: resolution = 70,000; AGC target = 3,000,000; maximum IT = 100 ms; scan range = 350 to 1800 *m*/*z*; (2) secondary mass spectrometry: resolution = 17,500; AGC target = 50,000; maximum IT = 120 ms; topN = 20; collision energy = 30.

### 2.10. Statistical Analysis

Data were calculated as mean ± standard deviation with 3 replicates. Significant differences were determined by one-way ANOVA followed by Duncan’s test (*p* < 0.05 was considered as statistically significant). IBM SPSS Statistics (version 26.0, Chicago, IL, USA) and GraphPad Prism (version 8.0, San Diego, CA, USA) were used to calculate and plot.

## 3. Results and Discussion

### 3.1. Effect of Storage on Peroxide Value

The PV indicates the content of lipid hydroperoxide in brown rice oil under the condition of self-oxidation and photo-oxidation, which reflects the oxidation degree of the sample [28]. Figure 1 shows the change in PV of Se-BR and BR during storage. The PV of the two kinds of brown rice first increased and then decreased. Specially, the PV of BR increased to the highest point of 1.232 mEq/kg from 0 day to 60 days, and then began to decline, indicating that the degradation rate of hydroperoxide in BR was higher than the formation rate after 60 days. As an intermediate product of lipid oxidation, hydroperoxide can be further degraded into secondary metabolites such as aldehydes and ketones [29]. The PV of Se-BR slowly increased to 1.115 mEq/kg from 0 day to 90 days and then tended to be flat. The upward and downward trend of the PV of BR was steeper than that of the PV of Se-BR. The increase in the PV of Se-BR was significantly lagging behind, which, to some extent, indicates that Se-BR has higher oxidation resistance stability than BR. Hence, selenium can effectively reduce the lipid oxidation rate during the storage of Se-BR.

### 3.2. Effect of Storage on Fatty Acid Value

The FAV is an indicator of the content of free fatty acids in oil, which reflects the quality change of brown rice during storage. As shown in Figure 2, the changes in FAV of the two kinds of brown rice were positively correlated with the storage time, indicating that the production rate of fatty acids was greater than that of the decomposition rate. The FAV of BR decreased from 150 days to 180 days, which may be related to the activity of enzymes that produce fatty acids [29]. Similar results were found in the study of Li et al. [30], demonstrating that selenium exerts a positive effect on the preservation of brown rice.

### 3.3. Effect of Storage on Carbonyl Value

Under the oxidative stress of free radicals, reactive oxygen species, nitrogen and other substances, the protein structure is easily changed, including amino acid side chain modification, polypeptide chain cross-linking and disulfide bond formation. Carbonyl derivatives are mainly produced through direct oxidation of amino acid side chains, cleavage of peptide bone chains and binding of other nonprotein carbonyl groups. Given the various mechanisms involved, carbonyl derivatives are widely considered one of the landmark products for describing protein oxidative damage [31]. Figure 3 shows the changes in CV of BR and Se-BR during storage. The CV of BR and Se-BR increased from 2.647 and 2.638 mEq/kg at 0 day to 8.571 and 8.0263 mEq/kg at 90 days, respectively, followed by a flattening, downward trend. The rate of increase in the CV of the two proteins is similar, which is the same as the results of Li et al. [11]. This finding indicates that the proteins in the two brown rice varieties were oxidatively damaged during storage.

### 3.4. Analysis of FTIR Spectra

FTIR is a commonly used method in studying the structure of proteins. The secondary structure of the two species of brown rice protein is shown in Table 1, from which the relative content of α-helix, β-sheet, β-turn and random coil changed significantly with the change in storage time. The β-sheet content of BRP first increased and then decreased, which might be due to the inversion structure of the β-turn of 180° to the extension structure of β-sheet, making the spatial conformation of the protein more stable. The increase in the random coil structure at 90 days of storage was insignificant, indicating that the α-helix also underwent a transition to a β-sheet structure, which may be related to the fact that, compared with other secondary structures, β-sheet is more sensitive to environmental changes [32]. However, with the deepening of the deterioration of brown rice, the three relatively ordered structures were destroyed, and the content of random coil increased significantly, demonstrating that the disordered structure in the protein increased and the ordered structure decreased, which may lead to a decrease in edible quality [33].

Different from BRP, the β-sheet structure of Se-BRP did not change significantly. The β-sheet gradually decreased during storage, whereas the content of random coil gradually increased, indicating the transformation of the secondary structure of the protein to a disordered one. The α-helix rose during 90–180 days of storage, implying that some kind of alteration in Se-BRP may promote an increase in ordered structure, thereby increasing the thermal stability of the protein to prevent further damage to the protein caused by storage [34]. This result suggests that, compared with BR, Se-BR has higher oxidative resistance stability of proteins.

### 3.5. Proteomic Analysis

To study the proteome changes in the two kinds of brown rice before and after storage, we identified a total of 2992 trusted proteins (confidence interval ≥ 95%). Credible DEPs were screened out (filter criteria: number of peptides ≥ 2; AVG ≥ 1.5 was selected as the upregulated protein and AVG ≤ 0.67 as the downregulated protein), and the statistical results of the DEPs were obtained, as shown in Table 2. Groups Se-0:Se-6 and BR-0:BR-6 identified 235 and 237 total DEPs, respectively, groups BR-0:Se-0 and BR-6:Se-6 identified 113 and 213 total DEPs, respectively. This result showed that both types of brown rice had different degrees of quality change during storage as the storage time increased. Through comparing the DEPs between groups, we can study the DEPs and metabolic pathways that play an important role in the storage quality of the two brown rice varieties under different storage conditions.

#### 3.5.1. Gene Ontology (GO) Analysis

GO analysis was performed of four groups of DEPs. Appendix A shows a bubble chart based on the number and significance of DEPs annotated to secondary classification items.

For groups Se-0:Se-6 and BR-0:BR-6, in terms of biological process (BP), the DEPs were mainly enriched in the negative regulation of hydrolase activity, cellular glucan metabolic process, negative regulation of molecular function, negative regulation of catalytic activity, negative regulation of proteolysis and negative regulation of peptidase activity. In terms of cellular component (CC), the DEPs were mainly enriched in amyloplast, aleurone grain, organelle outer membrane, vacuole and mitochondrial outer membrane. In terms of molecular function (MF), the DEPs were mainly enriched in nutrient reservoir activity, serine-type endopeptidase inhibitor activity, immunoglobulin binding and peptidase inhibitor activity.

For groups BR-0:Se-0 and BR-6:Se-6, in terms of BP, the DEPs were mainly enriched in the negative regulation of molecular function, negative regulation of catalytic activity, negative regulation of proteolysis, regulation of proteolysis, negative regulation of cellular protein metabolism process, regulation of cellular protein metabolism process and regulation of endopeptidase activity. In terms of CC, the DEPs were mainly enriched in extracellular regions, amyloplast, aleurone grain, vacuole, preribosome, intrinsic component of the chloroplast outer membrane and integral component of the plastid membrane. In terms of MF, the DEPs were mainly enriched in immunoglobulin binding, serine-type endopeptidase inhibitor activity, peptidase inhibitor activity, nutrient reservoir activity, endopeptidase regulator activity, molecular function regulator and antioxidant activity.

#### 3.5.2. Kyoto Encyclopedia of Genes and Genomes (KEGG) Pathway Analysis

In the Se-0:Se-6 and BR-0:BR-6 groups, the DEPs were mainly enriched in carbon fixation in photosynthetic organisms, pyruvate metabolism, starch and sucrose metabolism, peroxisomes, glutathione (GSH) metabolism and phenylpropanoid biosynthesis. In Se-0:Se-6, the DEPs were enriched in Cys and Met metabolism and sulfur metabolism pathways, which indicates that selenium may play a role in isotope substitution of sulfur in metabolism. Moreover, enrichment of the base excision repair and the zeatin biosynthesis pathway showed that the protein expression of Se-BR was more significant in reducing gene expression errors, reducing the protein solubilization rate and delaying seed aging. In BR-0:BR-6, the DEPs were enriched in phenylalanine, tyrosine and tryptophan biosynthesis, isoquinoline alkaloid biosynthesis, tropane, piperidine and pyridine alkaloid biosynthesis and ubiquinone and other terpenoid-quinone biosynthesis. Accordingly, under adverse conditions, BR initiates a certain defense mechanism to regulate its own information expression and cell signal transduction.

In the BR-0:Se-0 and BR-6:Se-6 groups, the DEPs were mainly enriched in carbon fixation in photosynthetic organisms, phenylpropanoid biosynthesis, base excision repair, starch and sucrose metabolism and phenylalanine, tyrosine and tryptophan biosynthesis. This result showed that, in addition to the primary metabolism and secondary metabolism, there were also differences in the regulatory effects of Se-BR and BR in information expression and antistress conditions under the same external conditions.

In terms of fatty acid synthesis and degradation, Os01g0919900 is a protein involved in the negative regulation of rice defense reactions, responsible for regulating fatty acid desaturase activity [35]. As shown in Table 3, the expression of this protein was upregulated in BR-0:BR-6 but showed no significant change in Se-0:Se-6, implying that BR had higher unsaturated fatty acid accumulation than Se-BR. 3-Ketoacyl-CoA thiolase-like protein plays an important role in fatty acid β-oxidative cleavage [36], expressing upregulation in both BR-0:BR-6 and Se-0:Se-6. 2-Hydroxyacyl-CoA lyase (2-HPCL) is a peroxisomal enzyme that depends on TPP to degrade the C-C bond of 3-methyl-branched fatty acids or shorten 2-hydroxy long-chain fatty acids [37]. The expression of 2-HPCL and ADH1 was upregulated in the BR-6: Se-6 group, indicating that Se-BR had higher expression of fatty acid degradation mechanism and lower lipid oxidation pressure on protein, which is consistent with the change trend of FAV measured previously.

Catalase catalyzes the decomposition of hydrogen peroxide into oxygen and water and reduces free radical stress. The expression of catalase was downregulated in BR-0:Se-0 and BR-0:BR-6 but upregulated in Se-0:Se-6 and BR-6:Se-6. Superoxide dismutase (SOD)[Mn] is an important antioxidant enzyme capable of catalyzing the disproportionation of superoxide anions and hydrogen peroxide [38]. The expression of SOD[Mn] was upregulated in Se-0:Se-6 and BR-6:Se-6, but no significant change was observed in BR-0:Se-0 and BR-0:BR-6. This result showed that Se-BR expressed more antioxidant enzyme activity than BR in a high-temperature storage environment, which is consistent with the previous experimental results; that is, Se-BR has stronger antioxidant stability. Glycolate oxidase 5 is a key enzyme in photorespiration metabolism and has been demonstrated as an alternative source for H_2_O_2_ production during gene-for-gene and non-host resistance in plants [39]. The expression of glycolate oxidase 5 was upregulated in BR-0:BR-6 but had no significant change in Se-0:Se-6, suggesting that regular brown rice has developed a source of H_2_O_2_ production independent of NADPH (nicotinamide adenine dinucleotide phosphate) oxidases and may be subjected to more oxidative stress than Se-rich proteins.

Glutathione transferase (GST) is one of the most important enzymes in biotransformation, which participates in cell anti-injury, anticarcinogenesis and other important metabolic processes [40,41]. The proteomic analysis showed that both kinds of brown rice underwent varying degrees of change: Q93WM2 was upregulated in BR-0:BR-6 but downregulated in BR-6:Se-6; Q945W6 was downregulated in BR-0:Se-0 and BR-0:BR-6; Q93WY5 was downregulated in BR-0:BR-6 but upregulated in Se-0:Se-6 and BR-6:Se-6; Q0JJ25 was upregulated in Se-0:Se-6. Meanwhile, the expression of L-ascorbate peroxidase was downregulated in both BR-0:BR-6 and Se-0:Se-6, suggesting that the two kinds of brown rice had protein antioxidant regulation to reduce the oxidation caused by high-temperature storage stress [42].

Peroxiredoxin (Prdx) and glutathione reductase (GR) (P48642, Q8S5T1) were upregulated in Se-0:Se-6 expression. Prdx is a peroxidase that detoxifies peroxides, such as hydrogen peroxide, has broad substrate specificity and plays a key role in protecting cells from stress and maintaining homeostasis in many cellular processes [43]. The active action of Prdx is based on conserved Cys residues and depends on intermolecular or intramolecular mercaptan dioxide as an electron donor regeneration cycle, e.g., GSH and thioredoxin [44]. Therefore, the upregulation expression in Se-BRP may be associated with the isotope substitution of selenium and sulfur. GSH has the functions of maintaining cell stability and preventing protein denaturation. With both oxidized and reduced forms, GSH is transformed into oxidized glutathione (GSSG) under oxidative stress and into GSH under the action of GR. Thus, GR is a key enzyme for maintaining the GSH/GSSG concentration ratio [45].

GR was upregulated in Se-GBP, indicating that, after selenium enrichment, GR was more sensitive to oxidative stress and expressed a stronger antioxidant effect. GSH can be metabolized to produce Cys and converted into Met, providing sulfur-containing amino acids for protein synthesis. Phosphoserine aminotransferase (PSAT), 5-methyltetrahydropteroyltriglutamate--homocysteine methyltransferase 2 and cysteine synthase (CS) [46,47] were upregulated in Se-0:Se-6, indicating that the sulfur metabolism in Se-BRP was faster. This finding is consistent with the higher GSH metabolism in Se-BRP; that is, Se-BR had higher levels of expression against oxidative stress.

Proteins associated with sugar metabolism, such as starch synthase, isoamylase 2, α- 1,4 glucan phosphorylase and malate dehydrogenase (MDH), were upregulated or downregulated in both brown rice proteins after 180 days of storage, suggesting that storage promoted glucose metabolism and accelerated the aging rate of brown rice [20]. Some enzymes involved in glucose metabolism are also involved in other biological metabolic processes. MDH is a key enzyme in the tricarboxylic acid (TCA) cycle; acetyl-CoA carboxylase (ACC) is involved in regulating the synthesis rate of fatty acids; pyruvate phosphate dikinase (PPDK) is a rate-limiting enzyme of the C4 pathway; hydroxyacylglutathione hydrolase (HAGH) is involved in the production of reductive glutathione. The different degrees of expression of these enzymes in the four groups of samples indicate that the protein expression of the two brown rice varieties involves distinct metabolic pathways. Figure 4 shows the differences in the expression of Se-BR and BR through TCA cycle, glycolysis/gluconeogenesis, GSH metabolism, starch metabolism and fatty acid metabolism after storage.

## 4. Conclusions

Lipid and protein oxidation occurred during the 180-day storage of Se-BR and BR. PV, FAV and CV showed that selenium-rich brown rice had higher lipid and protein oxidation resistance stability compared with ordinary brown rice; FTIR showed that storage led to the changes of protein secondary structure from ordered to disordered. By comparing the differences in protein expression before and after storage in BR and Se-BR, 235 DEPs were identified in group Se-0:Se-6, 237 DEPs were identified in group BR-0:BR-6, 113 DEPs were identified in group BR-0:Se-0 and 213 DEPs were identified in group BR-6:Se-6. The metabolic pathway of DEPs was analyzed by KEGG; it was found that storage promoted the glucose metabolism of the two kinds of brown rice and accelerated the aging rate of brown rice. The expression of catalase, SOD[Mn], GST, Prdx and GR was upregulated in Se-BRP, proving the oxidation resistance stability of Se-BR is higher than that of BR. The DEPs related to the synthesis and degradation of fatty acids show that Se-BR has higher efficiency in decomposing fatty acids and higher stability in lipid oxidation, which is consistent with the previous conclusions. This study provides insights for the storage and oxidation mechanism of ordinary brown rice and selenium-rich brown rice. On this basis, further research on the kinetics of protein oxidation induced by lipid oxidation products is helpful to control the quality and prolong the storage period of rice, which is of great significance for improving food quality and reducing resource waste.

## Figures and Tables

**Figure 1 foods-11-03878-f001:**
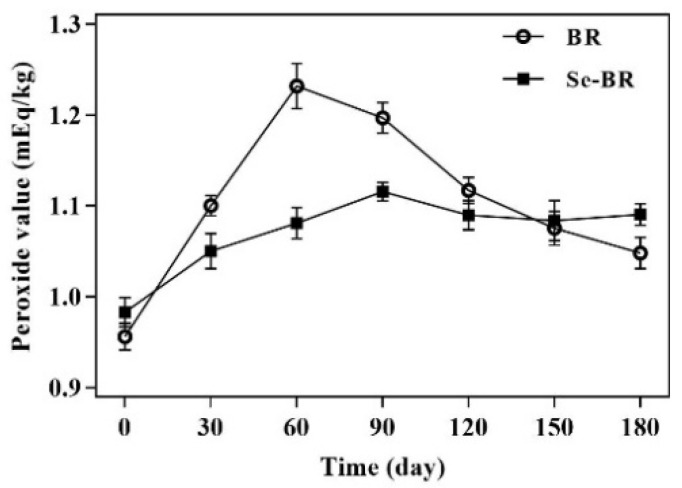
Peroxide value of ordinary brown rice (BR) and selenium-rich brown rice (Se-BR). Each data point is expressed as the mean and standard error of three replicates.

**Figure 2 foods-11-03878-f002:**
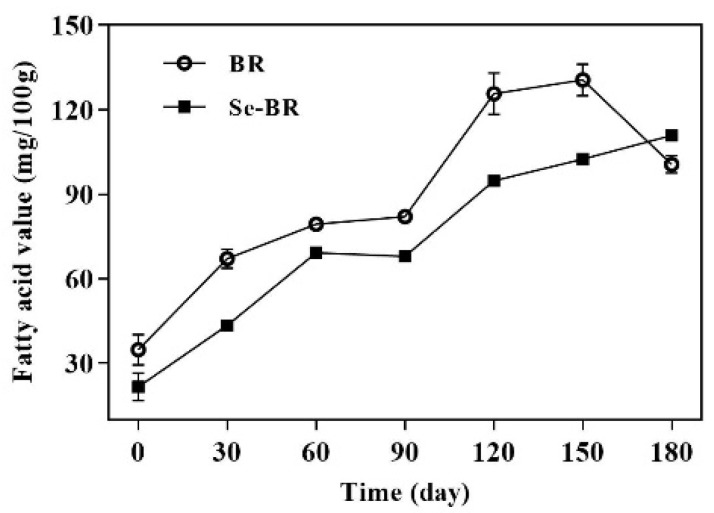
Fatty acid value of BR and Se-BR. Each data point is expressed as the mean and standard error of three replicates.

**Figure 3 foods-11-03878-f003:**
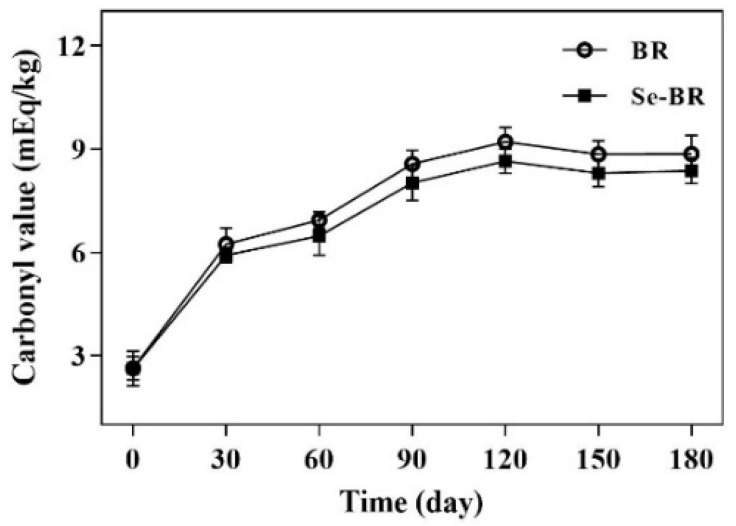
Carbonyl value of BR and Se-BR. Each data point is expressed as the mean and standard error of three replicates.

**Figure 4 foods-11-03878-f004:**
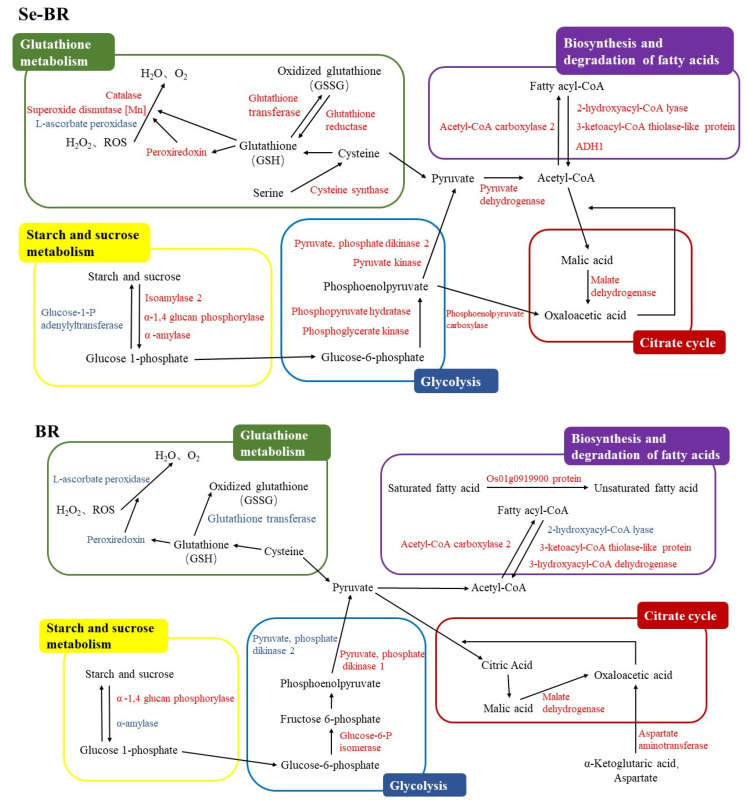
DEPs in the expression of Se-BR and BR in different metabolic pathways. Different metabolic pathways are distinguished by different colored boxes. Metabolic process substances are connected by arrows, and the direction of the arrow represents the direction of metabolism. Upregulated and downregulated proteins are marked in red and blue, respectively.

**Table 1 foods-11-03878-t001:** Secondary structure composition of BR and Se-BR.

Species	Time (d)	β-Sheet (%)	Random Coil (%)	α-Helix (%)	β-Turn (%)
BR	0	26.80 ± 0.26 ^b^	23.41 ± 0.18 ^a^	22.57 ± 0.02 ^b^	27.22 ± 0.06 ^b^
	90	29.37 ± 0.91 ^c^	24.10 ± 0.34 ^a^	21.51 ± 0.64 ^b^	25.02 ± 1.22 ^ab^
	180	23.74 ± 0.30 ^a^	34.41 ± 0.20 ^b^	17.11 ± 0.43 ^a^	24.73 ± 0.52 ^ab^
Se-BR	0	28.75 ± 0.07 ^a^	23.99 ± 0.07 ^a^	21.57 ± 0.09 ^a^	25.69 ± 0.09 ^b^
	90	28.99 ± 1.11 ^a^	24.50 ± 0.06 ^b^	21.06 ± 0.68 ^a^	25.45 ± 1.86 ^b^
	180	29.15 ± 0.29 ^a^	28.39 ± 0.09 ^c^	26.70 ± 0.21 ^b^	15.76 ± 0.16 ^a^

Different letters (a–c) of the same type in the same column indicate statistically significant differences (*p* < 0.05).

**Table 2 foods-11-03878-t002:** Statistics of different proteins (DEPs) identified by different groups.

Groups	Total Trusted Proteins	Total Differential Proteins	Upregulated Proteins	Downregulated Proteins
Se-0:Se-6	2992	235	167	68
BR-0:BR-6	2992	237	138	99
BR-0:Se-0	2992	113	48	65
BR-6:Se-6	2992	213	129	84

Se-0 refers to Se-BR that is stored for 0 day; Se-6 refers to Se-BR that is stored for 180 days; BR-0 refers to BR that is stored for 0 day; BR-6 refers to BR that is stored for 180 days.

**Table 3 foods-11-03878-t003:** Expression of DEPs by different metabolic pathways.

Citable Accession	No.	Name	BR-0:Se-0	BR-0:BR-6	Se-0:Se-6	BR-6:Se-6
**Peroxisome**						
B7EAG0	136	Catalase	0.6026	0.5248	1.8030	2.0512
Q6YT73	738	Glycolate oxidase 5	-	1.5996	-	-
Q43008	352	Superoxide dismutase [Mn], mitochondrial	-	-	4.6989	2.6546
**Glutathione metabolism**						
Q93WM2	1508	Glutathione transferase	-	2.8314	-	0.5445
Q945W6	2338	Glutathione transferase	0.6368	0.6607	-	-
Q93WY5	892	Glutathione transferase	-	0.6607	1.5417	2.0512
Q0JJ25	1175	Glutathione transferase	-	-	1.5417	-
B7ERQ1	145	Peroxiredoxin	0.4285	0.5200	2.0324	-
P48642	749	Glutathione reductase, cytosolic	-	-	1.5996	1.5136
A0A345YV75	1753	L-ascorbate peroxidase	-	0.6081	0.5058	-
Q8S5T1	1410	Glutathione reductase	-	-	1.5560	-
**Cysteine and methionine metabolism**						
Q8LMR0	317	Phosphoserine aminotransferase	-	-	1.9588	-
Q0JJ47	50	Aspartate aminotransferase	2.0512	2.5351	-	-
Q2QLY4	72	5-methyltetrahydropteroyltriglutamate-homocysteine methyltransferase 2	-	-	2.1677	1.5276
Q9XEA8	506	Cysteine synthase	-	-	5.5976	3.8019
**Fatty acid degradation**						
Q0JMH0	357	2-hydroxyacyl-CoA lyase	0.2992	0.4613	3.7325	2.4434
S4U008	178	3-hydroxyacyl-CoA dehydrogenase	-	1.7539	-	-
Q84P96	304	3-ketoacyl-CoA thiolase-like protein	-	1.7539	1.5704	-
D7PPK3	176	ADH1	-	-	20.7014	14.0605
**Biosynthesis of unsaturated fatty acids**						
A0A0N7KEB1	1635	Os01g0919900 protein	-	1.5136	-	-
**Starch and sucrose metabolism**						
D0TZR4	79	Starch synthase, chloroplastic/amyloplastic	2.0512	1.9409	0.5495	0.5861
R4JL77	51	Starch synthase, chloroplastic/amyloplastic (Fragment)	0.2109	0.2014	-	-
B7ESH5	2300	Starch synthase, chloroplastic/amyloplastic	0.3733	0.4875	2.0893	1.5849
Q69T99	402	Glucose-1-phosphate adenylyltransferase small subunit 1, chloroplastic/amyloplastic	-	0.5754	0.5546	-
B9EY77	32	Glucose-1-phosphate adenylyltransferase	1.5136	1.7061	-	0.6668
Q6AU80	984	Isoamylase 2, chloroplastic	-	-	46.1318	39.0841
Q8LQ33	510	Alpha-1,4 glucan phosphorylase	-	2.7290	6.9823	3.3113
B3IYE3	16	Alpha-1,4 glucan phosphorylase	-	2.1086	2.4889	-
Q0J528	520	Alpha-amylase	0.4529	0.3981	2.9648	3.2810
Q0J136	123	Glucose-6-phosphate isomerase	-	1.7378	-	-
D0TZK6	1186	1,4-alpha-glucan branching enzyme	-	0.5445	-	-
Q0D9D0	9	1,4-alpha-glucan branching enzyme	-	2.7542	-	0.4207
Q0J8G4	448	Fructokinase-2	-	-	1.5276	-
B9FUP9	737	Amylomaltase	-	-	1.5417	-
Q653V7	33	Probable alpha-glucosidase Os06g0675700	-	-	1.7539	1.9055
Q93 × 08	26	UTP--glucose-1-phosphate uridylyltransferase	-	-	1.6444	1.5276
A3AC56	2331	Trehalose 6-phosphate phosphatase	-	-	-	1.8030
**Pyruvate metabolism**						
Q42972	536	Malate dehydrogenase, glyoxysomal	0.6368	0.4875	1.5417	1.9588
A1YQK1	125	Malate dehydrogenase	-	1.6144	1.5136	-
Q6AVA8	10	Pyruvate, phosphate dikinase 1, chloroplastic	-	3.0761	-	0.3873
Q75KR1	113	Pyruvate, phosphate dikinase 2	-	0.6026	1.5849	1.9588
Q65WW3	488	Os05g0474600 protein	-	1.9055	2.9648	1.7219
B9FK36	67	Acetyl-CoA carboxylase 2	-	3.1333	1.7865	0.6368
Q10LW8	361	Hydroxyacylglutathione hydrolase, putative, expressed	-	0.5808	-	1.5417
Q7XKB5	203	Pyruvate kinase	-	-	3.3729	3.6308
Q9FYX8	362	Phosphoenolpyruvate carboxylase	0.6310	-	2.8054	2.2909
D7PPK3	176	ADH1	-	-	20.7014	14.0605
Q6Z5N4	1472	Pyruvate dehydrogenase E1 component subunit alpha-1, mitochondrial	-	-	3.7325	4.7424
Q7XTJ3	974	Pyruvate dehydrogenase E1 component subunit alpha-3, chloroplastic	-	-		1.9055
Q7XAL3	1500	Acetyltransferase component of pyruvate dehydrogenase complex	1.6293	-	0.5861	-
Q5JKW5	177	Malic enzyme	-	-	1.6904	-
A3BQD6	101	Glyoxalase I	-	-	-	1.6444
**Glycolysis/Gluconeogenesis**						
Q655T1	294	Phosphoglycerate kinase	-	-	3.6983	3.2211
A1YQJ3	54	Phosphopyruvate hydratase	-	-	1.7378	1.7219
B9F7T1	205	Pyruvate decarboxylase	-	-	2.3550	1.9953
Q84TX6	361	Hydroxyacylglutathione hydrolase, putative, expressed	-	0.5808	-	1.5417
**Citrate cycle (TCA cycle)**						
Q6ZL94	684	Succinate-CoA ligase [ADP-forming] subunit alpha	-	1.5276	-	-
Q7XVM2	442	Dihydrolipoyllysine-residue succinyltransferase	-	-	1.5704	-

## Data Availability

Data is contained within the article or Appendix A.

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
