# Peer review of "Lipid and Protein Oxidation of Brown Rice and Selenium-Rich Brown Rice during Storage"

_foods, 2022, doi:10.3390/foods11233878_

Round 1
Reviewer 1 Report
- Please rewrite and organize the abstract according to the following context:
A short introduction, hypothesis (aim) of the study, methods, the most important quantitative results, a general conclusion, and future prospective
- Page 2 line 70, please state the city and country
- Most of the methods were mentioned in brief, please reformat it in more detail
- In preparation of brown rice protein, please add this method in detail and state a reference
- Page 2 line 81, replace was to were
- Page 2 line 84, replace were to was
- Please modify, PV were determined to PV was ...
- Fourier transformed infrared (FTTR), modify to (FTIR). Please state this method in detail
- Fig 4 should move to supplementary material
- Conclusions section, please highlight the future standpoint well.
- Manuscript has grammatical errors, please check.
Author Response
Please rewrite and organize the abstract according to the following context: a short introduction, hypothesis (aim) of the study, methods, the most important quantitative results, a general conclusion, and future prospective.
Response: We would like to thank you very much for your comments. We have revised the abstract according to your comments.
Page 2 line 70, please state the city and country.
Response: The suggested change has been made. In addition, we have modified “Henan Jinbasu Rice Industry Co., LTD” to “Yuanyang Basu Rice Industry Co., LTD”.
Most of the methods were mentioned in brief, please reformat it in more detail.
Response: The suggested change has been made. We have added more operational details in the methods of “2.7. Determination of carbonyl value” and “2.8. Fourier transformed infrared (FTIR) spectroscopy”.
In preparation of brown rice protein, please add this method in detail and state a reference
Response: The suggested change has been made. We have cited corresponding references.
Page 2 line 81, replace was to were.
Response: The suggested change has been made.
Page 2 line 84, replace were to was.
Response: The suggested change has been made.
Please modify, PV were determined to PV was ...
Response: The suggested change has been made.
Fourier transformed infrared (FTTR), modify to (FTIR). Please state this method in detail.
Response: The suggested change has been made. We have added more operational details.
Fig 4 should move to supplementary material
Response: The suggested change has been made.
Conclusions section, please highlight the future standpoint well.
Response: The suggested change has been made. Since this research is theoretical, we have proposed possible future dynamic studies and practical applications on this basis.
Manuscript has grammatical errors, please check
Response: The suggested change has been made. We have checked the manuscript and corrected some errors.
Reviewer 2 Report
1- In section 2.2, isn't storage at 35 °C too high? The authors should explain the reasons for choosing this temperature value. Most products cannot withstand this temperature. Storage should also be done at lower temperatures for comparison.
2- Since it is a storage study, the seed moisture content must be given.
Author Response
In section 2.2, isn't storage at 35 °C too high? The authors should explain the reasons for choosing this temperature value. Most products cannot withstand this temperature. Storage should also be done at lower temperatures for comparison.
Response: We would like to thank you very much for your comments. The lipid and protein content in rice is low, and the experimental period of simulated rice storage in the laboratory environment is long, so we use a relatively high storage temperature to accelerate the deterioration process of rice storage.
Since it is a storage study, the seed moisture content must be given.
Response: The suggested change has been made.
Reviewer 3 Report
In abstract practical application is missing
Revise the list of keywords and avoid those keywords which are already in manuscript title
do not use too much short forms in abstract
there is technical writing mistake like use of , etc
correct for grammatically mistakes also
introduction is too short
Materials and methods headings and subheadings are too short, they need more explanations
add more relevant pictures and figures
reduce fig caption length and make short but more explanation add into the text
Fig 5 flow of direction and info not clear, improve
Try to avoid using short forms in headings and subheadings
page 5, line 190 .... FTIR is a commonly used method in studying the structure of proteins.
no need of such common info, discuss ur results and check same in whole manuscript
Table 2, put proper headings
use most relevant and latest references
Author Response
In abstract practical application is missing.
Response: We would like to thank you very much for your comments. As it is a preliminary study, this research is theoretical, which will provide theoretical basis for future application in practice.
Revise the list of keywords and avoid those keywords which are already in manuscript title
Response: The suggested change has been made. We have added storage protein, sulfur metabolism, and oxidation resistance as new keywords.
Do not use too much short forms in abstract
Response: The suggested change has been made.
There is technical writing mistake like use of , etc. Correct for grammatically mistakes also.
Response: The suggested change has been made. We have rechecked the manuscript writing and corrected some errors.
Introduction is too short
Response: The suggested change has been made. We have added some details of protein oxidation and proteomics.
Materials and methods headings and subheadings are too short, they need more explanations
Response: The suggested change has been made. We have modified some titles to make them look more complete.
Add more relevant pictures and figures
Response: The suggested change has been made. We have added the moisture content of the sample.
Reduce fig caption length and make short but more explanation add into the text
Response: The suggested change has been made.
Fig 5 flow of direction and info not clear, improve
Response: The suggested change has been made. We have added descriptions of the boxes and arrows in the diagram.
Try to avoid using short forms in headings and subheadings
Response: The suggested change has been made.
Page 5, line 190 .... FTIR is a commonly used method in studying the structure of proteins. No need of such common info, discuss ur results and check same in whole manuscript
Response: The suggested change has been made. We have removed some of the common information about FTIR and checked the manuscript.
Table 2, put proper headings
Response: The suggested change has been made. The original title was changed to “Statistics of different proteins (DEPs) identified by different groups”.
Use most relevant and latest references
Response: The suggested change has been made. We have updated some of the references.
Round 2
Reviewer 1 Report
The MS has been significantly improved and I appreciate the authors' efforts to respond to reviewer comments. So, the MS can be published in its current form.
Reviewer 2 Report
Necessary corrections and suggestions were made by the authors.